# High-Frequency Pulsed Electric Field Ablation in Beagle Model for Treatment of Prostate Cancer

**DOI:** 10.3390/cancers14204987

**Published:** 2022-10-12

**Authors:** Seung Jeong, Song Hee Kim, Hongbae Kim, Jeon Min Kang, Yubeen Park, Dong-Sung Won, Ji Won Kim, Dae Sung Ryu, Chu Hui Zeng, Jong Hoon Chung, Bumjin Lim, Jung-Hoon Park

**Affiliations:** 1Department of Biosystems & Biomaterials Science and Engineering, Seoul National University, Seoul 08826, Korea; 2Biomedical Engineering Research Center, Asan Institute for Life Sciences, Asan Medical Center, 88 Olympic-ro 43-gil, Songpa-gu, Seoul 05505, Korea; 3Research Institute for Agriculture and Life Sciences, Seoul National University, Seoul 08826, Korea; 4Department of Urology, Asan Medical Center, University of Ulsan College of Medicine, 88 Olympic-ro 43-gil, Songpa-gu, Seoul 05505, Korea

**Keywords:** irreversible electroporation, high-frequency pulsed electric field, prostate cancer, decellularization, regeneration

## Abstract

**Simple Summary:**

Around 248,530 newly diagnosed in 2021, prostate cancer has been the most frequently diagnosed cancer in the USA. Rapid decellularization preserving acellular tissues is essential for accurate treatment and regeneration in affected areas. Our study aimed to assess the safety and efficacy of high-frequency pulsed electric field (HF-PEF) in a beagle model for the treatment of prostate cancer. We confirmed that HF-PEF of 1800 V/cm, 100 μs width, 2 ms interval successfully decellularized the prostate tissue after 4 hours, and the tissues were almost regenerated in 28 days. IRE with HF-PEF has therapeutic potential to treat prostate cancer while minimizing damage to the surrounding tissues.

**Abstract:**

Conventional irreversible electroporation (IRE) with low-frequency pulsed electric field (LF-PEF) is used to induce cell death; however, it has several disadvantages including a long procedure time and severe muscle contraction due to high-voltage electric field. This study investigates a novel IRE protocol with high-frequency pulsed electric field (HF-PEF) of 500 Hz repetition to ablate the prostate tissue in beagles for treatment of prostate cancer. A finite element analysis was performed to validate optimal electrical field strength for the procedure. In total, 12 beagles received HF-PEF of 500 Hz and were sacrificed at 4 h, 4 days, and 28 days (3 each). The remaining three beagles underwent sham procedure. The outcomes of HF-PEF were assessed by histological responses. HF-PEF successfully decellularized the prostate tissues 4 h after the treatment. The prostate glands, duct, and urethra were well preserved after IRE with HF-PEF. The ablated prostatic tissues were gradually regenerated and appeared similar to the original tissues 28 d after IRE with HF-PEF. Moreover, electrocardiography and hematology demonstrated that IRE with HF-PEF did not seriously affect the cardiac tissue. HF-PEF was effective and safe in the beagle prostate and effectively induced the ablation and gradually recovered with cellular regeneration.

## 1. Introduction

Prostate cancer is the most frequently diagnosed cancer in the USA, with approximately 248,530 cases in 2021 [1]. Radical prostatectomy is a standard therapeutic option to improve survival rates and delay the metastatic progression of locally advanced prostate cancer (LAPRC) [2,3]. However, the standard treatment has adverse effects such as erectile dysfunction and urinary incontinence, thus reducing quality of life [4,5,6]. Several localized therapeutic modalities have been developed to spare the adjacent vessels or neurovascular bundles near the targeted sites in the prostate [7]. Focal therapies for prostate cancer aim to achieve oncologic control, while reducing the adverse effects of whole-gland therapies [8].

Among various techniques for localized tumor ablation, irreversible electroporation (IRE) is a non-thermal ablation sparing extracellular matrix, vasculature, nerves, and ductal networks, which causes rapid tissue regeneration and tissue function preservation [9]. Most studies on IRE focused on low-frequency pulsed electric field (LF-PEF) at approximately 1 Hz pulse repetition rate [10,11,12]. However, LF-PEF has several disadvantages, including long procedure time and muscle contraction owing to high-voltage electric field, causing unpleasant sensations [13,14]. Various frequencies from 1 Hz to 2.5 kHz provided different apoptotic rates [13]. Possible mechanisms for frequency parameter of PEF are based on the high-frequency pass filter of the cellular membrane [15] and transient rise of Ca^2+^ concentration in the cytosol [16]. In the recent in vitro study, nanoseconds PEF (nsPEF) of 1 MHz frequency significantly enhanced cytotoxic effect of the prostate cancer cells in the presence of extracellular Ca^2+^ [17]. Despite of the enhanced cytotoxic effect in nsPEF, PEF of 1 kHz is estimated to cause less temperature rise than nsPEF, which causes thermal damage in the tissues [18,19]. Meanwhile, High-frequency IRE (H-FIRE), biphasic pulses of 90~140 kHz, was designed to decellularize beagle prostate tissue for prostate cancer ablation [20]. However, relatively high voltages ranging 3500~5000 V were required to cause sufficient ablation area for H-FIRE. In this study, High-frequency PEF (HF-PEF) with monophasic pulses of 1.8 kV, 500 Hz was designed to ablate beagle prostate tissue. Since very few studies have investigated beagle prostate tissue decellularization and regeneration after IRE with HF-PEF, this study aimed to investigate the safety and efficacy of IRE with HF-PEF in beagle prostate using monopolar electrodes.

## 2. Materials and Methods

### 2.1. HF-PEF System and Electrodes

The EPO-IRE^®^ system (EPO-S1; The Standard Co., Ltd.; Gunpo-si, Korea), with two parallel 15 mm exposed and 10 mm apart 19 G monopolar electrodes (Smart EPO Probe; The Standard Co., Ltd.), was utilized to deliver HF-PEF to the beagle prostate (Figure 1a). The system set an electric field intensity up to 3000 V with 100–1000 µs pulse widths at 100–2000 µs pulse intervals.

### 2.2. Simulation of Electrical Field and Temperature Distribution

A finite element analysis-based model was adopted using COMSOL Multiphysics 5.6 (Stockholm, Sweden) to simulate the electrical field strength and thermal distribution of HF-PEF. Two monopolar electrodes were used to apply 10 pulses of 1800 V, 100 μs width, and 2 ms delay. The thermal and electrical properties of the prostate tissue were determined from the online database (https://itis.swiss/virtual-population/tissue-properties/database/, accessed on 9 February 2021). A stationary study solver was adopted for electric field strength evaluation, while a time-dependent study solver was used for thermal distribution around the monopolar electrodes in the prostate.

### 2.3. Animal Study Design

This study was approved by the Institutional Animal Care and Use Committee of the Asan Institute for Life Sciences (IACUC-2020-14-277) and conformed to the US National Institutes of Health guidelines for handling laboratory animals. The study was conducted in compliance with the ARRIVE guidelines. After acclimatizing for 7 d, 12 certified healthy 12-month-old 14.5–16.2 kg (median weight: 15.35 kg) male beagle dogs (JA BIO, Suwon, Korea) were used for this study. Subsequently, the animals were maintained in individual >0.74 m^2^ cages at 23 ± 2 °C, 50 ± 10% relative humidity, 10–15 times/h ventilation frequency, and 12/12 h light/dark cycle. Then, three beagles were used as the control for presenting the normal prostate values, and the remaining nine beagles underwent IRE with HF-PEF using two monopolar electrodes. The nine beagles were randomly euthanized 4 h (*n* = 3), 7 d (*n* = 3), and 28 d (*n* = 3) after the procedure by intravenously injecting potassium chloride (KCl; DAI HAN PHARM CO., Seoul, Korea).

### 2.4. IRE with HF-PEF in the Beagle Prostate

All beagles were anesthetized using a mixture of 50 mg/kg zolazepam, 50 mg/kg tiletamine (Zoletil 50; Virbac, Carros, France), and 10 mg/kg xylazine (Rompun; Bayer HealthCare, Leverkusen, Germany). An endotracheal tube was then placed, and anesthesia was administered by inhalation of 0.5–2% isoflurane (Ifran^®^; Hana Pharm. Co., Seoul, Korea) with 1:1 oxygen (510 mL/kg per min). The techniques of the IRE with HF-PEF has been described in detail previously [21]. Briefly, a 70 mm midline incision was made on the abdomen of each anesthetized dog from the xiphoid process down to expose the urinary bladder and urethra. The prostate was then identified by tracing along the urinary tract. Two monopolar electrodes with 3D-printed spacer of 10 mm distance were penetrated the randomized side of the prostate under ultrasound guidance (Figure 1b,c). The HF-PEF used in a previous study was used: electric field strength, 1800 V/cm; pulse width, 100 μs; and pulse interval, 2 ms [22]. A total of 90 pulses were divided into 9 cycles of 10 pulses each to avoid thermal damage from intense electrical current. A digital oscilloscope (TDA3044B, Tektronix, Beaverton, OR, USA) was utilized to measure the electric current, and a hole-type current probe (TCP305A, Tektronix) was clamped to a cord to connect the pulse generator and the electrodes. The conductivity of the beagle prostate was also measured before, during, and after IRE with HF-PEF using the 4192A LF Impedance Analyzer (Yokogawa-Hewlett-Packard Ltd., Hachioji-Shi, Japan). After the procedure, the abdomen was irrigated with 0.9% saline, and the incision was closed with suture. Body temperature, appetite, pain, fecal output, and other clinical courses were monitored for one week, and antibiotics (gentamicin, 80 mg/2 mL; SHIN POONG PHARM Ltd., Seoul, Korea) and analgesics (keromin, Ketorolac 30 mg; HANA PHARM Ltd., Seoul, Korea) were routinely used for 3 d after the procedure.

### 2.5. Electrocardiography

The electrocardiography (ECG) signals were monitored by the Ag/AgCl electrodes (2223H, 3M; Saint Paul, MN, USA) connected with an ECG amplifier (PSL-iECG, PhysioLab; Busan, Korea). The fur above the lower third of the humerus and femur was shaved, and the biomedical sensor pads were attached to the sole and skin. The signals were measured to investigate the cardiac safety when HF-PEF was performed. ECG and heart rate were continuously monitored and recorded from 30 s before to 230 s after the procedure.

### 2.6. Hematological Examination

Blood serum samples were obtained immediately before and after, and 1 and 7 d after IRE procedure to measure cardiac troponin I and creatine kinase-MB (CK-MB) fraction levels to monitor cardiac function and inflammatory response. Approximately 3 mL blood was collected and transferred into Vacutainer SST Tubes. The tubes were pre-chilled, and samples were centrifuged at 3500 rpm for 10 min at 4 °C. Then, the serum was transferred to polypropylene tubes and stored at −80 °C for further analysis.

### 2.7. Gross and Histological Examinations

Gross and histopathological appearances of HF-PEF-induced wounds of the extracted urethra, prostate, and urinary bladder were evaluated. After being gently separated from the urethra and bladder, the prostate was fixed in 10% neutral buffered formalin for at least 24 h. Then, the HF-PEF-ablated prostate was sectioned transversely into proximal, middle, and distal portions for gross and microscopic examinations. The slides were stained with hematoxylin and eosin (H&E) and Masson’s trichrome (MT) for histological analysis. The degree of inflammatory cell infiltration was revealed by H&E staining and subjectively classified into 1, mild; 2, mild to moderate; 3, moderate; 4, moderate to severe; and 5, severe, according to inflammatory cell distribution. Digitalized slides were viewed using a scanner (Pannoramic 250 FLASH III, 3D HISTECH Ltd.; Budapest, Hungary) and analyzed using its viewer software (CaseViewer, 3D HISTECH Ltd.). The MT fibrotic areas were evaluated using the ImageJ 1.53c software (Wayne Ribband, National Institutes of Health; Bethesda, MD, USA). Trainable Weka Segmentation was performed to create a binary image of MT-stained ablated area. Analyses of the histological findings were based on the consensus of three observers blinded to the experimental groups.

### 2.8. Immunohistochemistry

Immunohistochemistry (IHC) was performed on paraffin-embedded sections using terminal deoxynucleotidyl transferase-mediated dUTP nick and labeling (TUNEL; S7100, ApopTag Peroxi-199 dase, Sigma Aldrich; St. Louis, MO, USA) primary antibodies to verify apoptosis after the HF-PEF treatment. The sections were visualized using a stainer (BenchMark XT, Ventana Medical Systems, Tucson, AZ, USA). The ImageJ 1.53c software was used to measure the ablation area stained with TUNEL. Trainable Weka Segmentation was performed to create a binary image of TUNEL-stained ablated area.

### 2.9. Immunofluorescence

Regeneration by cell proliferation was assessed by immunofluorescence (IF) staining of Ki-67 (Thermo Fisher Scientific, Waltham, MA, USA). Briefly, the fixed samples were sliced to 3 µm thickness, dehydrated with ethanol, and washed with tap water at room temperature. The slides were incubated in 3% H_2_O_2_ for 10 min to block endogenous peroxidase activity, 10% normal serum albumin for 30 min to block non-specific immunoglobulin binding, and primary antibody diluted in phosphate-buffered saline (PBS) for 60 min. After washing the slides in PBS with 0.5% Tween for 5 min, the slides were incubated with Alexa Fluor 594-conjugated Ki67 secondary antibody (1:200, Thermo Fisher Scientific) for 60 min and the cell nuclei were stained with mounting medium with DAPI (50011, Ibidi, Gräfelfing; Germany) for 1 min at room temperature. The percentage of Ki-67 positive area was calculated as 100% × (number of positive cells in the ablated area/total number of cells in the ablated area).

### 2.10. Statistical Analysis

Data were expressed as mean ± standard deviation (SD). Differences between the groups were analyzed using two-sample *t*-test and Mann–Whitney U test, as appropriate. *p* < 0.05 was considered statistically significant. Statistical analyses were performed using the SPSS software (version 27, IBM; Chicago, IL, USA).

## 3. Results

### 3.1. Simulated Electric Field and Temperature Distribution

When 1800 V/cm was applied, the cross-sectional and longitudinal electric field between the two electrodes were oval and dumbbell-shaped, respectively. Simulation of thermal distribution at 1800 V/cm showed a slight increased temperature at the electrode periphery. The maximum temperature between the electrodes was 37.4 ℃ at 1800 V/cm (Figure 2).

### 3.2. Procedural Outcomes

The IRE with HF-PEF procedures were technically successful in all beagles. All beagles showed mild hematuria and erection with stiff surrounding prostate tissue immediately after the procedure but resolved spontaneously within 7 days. All enrolled beagles survived until the end of the study without procedure-related death.

The IRE with HF-PEF pulses were successfully delivered to the prostate. The maximum increase in current was approximately 10 A (Figure 3a). The electrical conductivity of the prostate tissue significantly increased after IRE with that HF-PEF compared with before the procedure (2282.44 ± 96.21 μS/cm vs. 607.73 ± 47.51 μS/cm, *p* < 0.001) (Figure 3b).

### 3.3. Cardiac Safety after IRE with HF-PEF

ECG of the beagles did not interfere with after the procedure. During the procedure, mild muscle contraction was observed in all dogs, but the ECG showed a rapid heart rate drop in all dogs immediately after the procedure. No abnormality in the ECG was noted before and after the procedure (Figure 3c).

Cardiac troponin I and CK-MB levels were maintained during and after IRE with HF-PEF. Moreover, the cardiac troponin I (pre, post, 1 d, and 7 d: 156.6 ± 39.724, 161.0 ± 44.211, 119.2 ± 22.4, and 89.0 ± 40.6 pg/mL, respectively) or CK-MB (pre, post, 1 d, 7 d: 10.6 ± 0.1, 6.2 ± 4.2, 9.6 ± 1.4, and 12.9 ± 0.5 ng/mL, respectively) levels did not differ significantly between the groups (Figure 3d,e).

### 3.4. Gross and Histological Findings

Gross findings are shown in Figure 4. The ablated areas of the beagle prostate after IRE with HF-PEF maximized 4 h after the treatment and gradually decreased over time. These findings were similar to the proximal, middle, and distal cross-sections of the prostate. Histological findings are summarized in Appendix A and examples are shown in Appendix A and Figure 5 and Figure 6. The mean (±SD) degree of inflammatory cell infiltration did not differ 4 h after IRE treatment from that in the control (1.40 ± 0.52 vs. 1.11 ± 0.33, *p* = 0.164). However, the mean (±SD) degree of inflammatory cell infiltration significantly increased 7 d (4.00 ± 1.12, *p* < 0.05) and 28 d (2.80 ± 0.60, *p* < 0.05) after IRE treatment from that in the control. The mean (±SD) Masson’s trichrome (MT) fibrotic areas significantly increased 4 h after IRE treatment from that in the control (66.75 ± 14.03 mm^2^ vs. 21.64 ± 0.58 mm^2^, *p* < 0.05). Subsequently, the areas gradually decreased 7 d (32.04 ± 5.10 mm^2^, *p* = 0.070) and 28 d (30.31 ± 14.34 mm^2^, *p* = 0.145) after IRE.

### 3.5. Immunohistochemical Findings

The mean (±SD) TUNEL-stained ablation area was significantly increased after 4 h IRE treatment compared with that in the control (113.53 ± 5.57 mm^2^ vs. 2.69 ± 0.51 mm^2^, *p* < 0.01). Then, the areas were decreased at 7 d (41.65 ± 5.99 mm^2^, *p* < 0.05) and 28 d (4.46 ± 1.20 mm^2^, *p* = 0.063) after IRE with HF-PEF (Figure 6).

### 3.6. Immunofluorescence Findings

The percentage of Ki-67 positive area significantly decreased 4 h after IRE treatment compared with that in the control (0.14 ± 0.22% vs. 0.70 ± 0.67%, *p* < 0.05). Subsequently, the percentage increased 7 d (0.51 ± 0.13%, *p* = 0.09) and 28 d (0.57 ± 0.07%, *p* = 0.082) after IRE with HF-PEF, which was similar to that of the control (Figure 6).

Ki-67 positive areas were almost absent around the urethra 4 h after IRE treatment, while it maximized around the urethra 7 d after the procedure. Ki-67 positive areas almost disappeared 28 d after IRE with HF-PEF, which was similar to that in the control (Appendix A). The distance between the prostate glands widened with decreased Ki-67 detection around the glands 4 h after IRE with HF-PEF. Moreover, the glands were distorted and shrunk 7 d after IRE with HF-PEF. However, the prostate glands almost recovered 28 d after IRE with HF-PEF, which was similar to that in the control (Figure 5c).

## 4. Discussion

The IRE with PEF has many advantages to treat LAPRC by sparing the extracellular matrix, such as the neurovascular bundles, ducts, and major vasculatures in the target tissues, rapidly regenerating the tissues [23]. In the recent, H-FIRE consisting of several bi-phasic pulses was proposed for prostate cancer ablation to reduce muscle contraction and prevent arrhythmias [20]. However, relatively high voltages ranging 3500~5000 V were required to cause sufficient ablation area for H-FIRE. Moreover, arrhythmias are less likely to occur because the prostate is located away from the heart [24]. Despite previous studies on IRE with PEF, no studies have investigated the efficacy of IRE with HF-PEF consisting of several monophasic pulses in the beagle prostate, except for one study with a short period of observation [21]. A previous study has reported that the prostate glands were swollen for 18 h after IRE with HF-PEF without information on the histological changes or regeneration. Moreover, it applied 700 V, which is not enough to ablate sufficient prostate tissue volume [21].

HF-PEF of 1800 V was delivered in 9 groups of 10 pulses each (total 90 pulses) using two monopolar electrodes according to the electrical condition reported previously [25]. Monopolar electrodes were inserted under ultrasound guidance parallelly using a 3D-printed spacer. The animals were euthanized 4 h, 7 d, and 28 d after IRE with HF-PEF to investigate the histomorphological changes, cellular regeneration, and surrounding tissue damages.

Our study demonstrated that IRE with HF-PEF sufficiently ablated the prostate through increased current and electrical conductivity. A previous study show that electrical conductivity in the prostate increased from 0.284 to 0.927 S/m, which is 3.26-fold higher than that required to sufficiently ablate the targeted area [25]. In this study, electrical conductivity in the prostate increased from 0.607 to 2.282 S/m, which is 3.76-fold higher than that required to sufficiently ablate the area. Therefore, the energy level of HF-PEF was high enough to induce cell death of the prostate tissues.

Cardiac safety after IRE with HF-PEF was investigated through ECG and cardiac markers, Troponin I and CK-MB, in this study. Troponin I level of the IRE-treated beagles was below 0.2 ng/mL and did not increase over time, which was consistent with that reported previously [26]. ECG was distorted during the IRE procedure but restored its original shapes and frequencies instantly. CK-MB level in the IRE-treated beagles did not significantly increase after IRE with HF-PEF compared with that in the control beagles. Overall, the cardiac safety of IRE with HF-PEF was verified in beagles.

Apoptosis in the ablated area following HF-PEF was investigated by TUNEL assay and hematoxylin and eosin (H&E) staining. Gross examination revealed that histological changes around the electrodes were maximized at 4 h after IRE with HF-PEF and decreased over time, which was consistent with the results of TUNEL assay and H&E staining. In addition, TUNEL-positive cells in the ablated area indicated a significant increase in apoptosis, while the urethra, vessels, ducts, and gland structures were well preserved.

The degree of cellular regeneration of the ablated prostate tissue was investigated by Ki-67 immunohistochemistry (IHC) and MT staining. The urothelium cells around the urethra experienced apoptosis 4 h after IRE with HF-PEF. Newly proliferating urothelium cells were detected 7 d after the treatment and urethral structures were restored 28 d after the treatment. Prostate gland structures were maintained but fibrotic changes were observed at the increased interstitial glands 4 h after the treatment. The newly proliferating cells gradually increased along the border of the glands over time. The gland structures almost restored their original shapes, and the fibrotic area reduced 28 d after IRE with HF-PEF. The prostate glands in the IRE-treated beagles gradually regenerated 28 d after IRE with HF-PEF and appeared similar to those in the controls.

The study had several limitations. First, IRE with HF-PEF was evaluated in animals with normal prostate, so wound healing after IRE-induced injuries may differ considerably from a disease model. Second, the 28 d follow-up may not be long enough to monitor regeneration after IRE procedure in the beagle prostate. Furthermore, only a single electrical field strength (1800 V/cm) was tested; thus, a dose range study should be further performed for evaluating its safety and efficacy. Although additional studies are needed, the results of our study supported the basic concept of IRE with HF-PEF as localized ablation for the beagle prostate.

## 5. Conclusions

In summary, IRE with 1800 V/cm HF-PEF successfully decellularized the beagle prostate. The urethra, prostate glands, ducts, and vessels were well preserved, and the treated areas gradually regenerated 28 d post procedure. Our results demonstrated that focal treatment with IRE using HF-PEF is safe and effective in the beagle prostate. IRE with HF-PEF has therapeutic potential to treat LAPRC while minimizing damage to the surrounding tissues of the prostate.

## Figures and Tables

**Figure 1 cancers-14-04987-f001:**
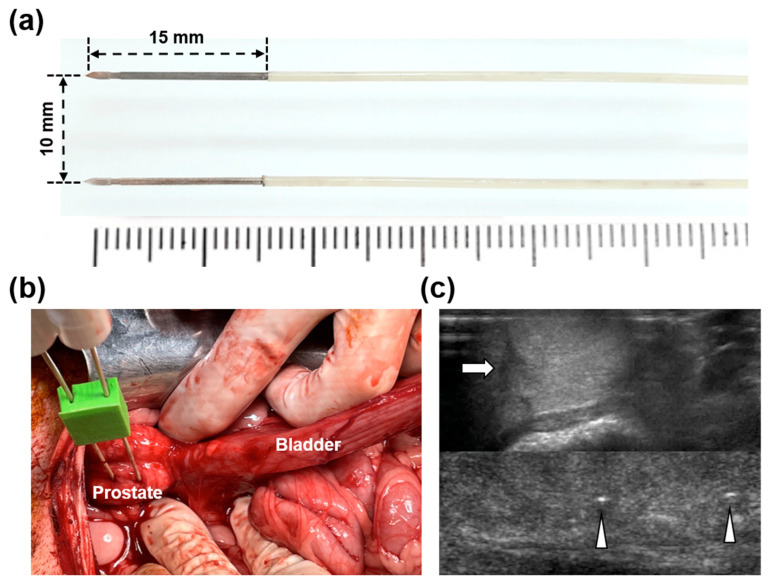
Monopolar electrodes and procedural steps of irreversible electroporation with high-frequency pulsed electric field in the beagle prostate. (**a**) Photograph showing stainless steel monopolar electrodes with 15 mm exposed and 10 mm between the electrodes. (**b**) Photograph obtained during the procedure showing the bladder, prostate, and inserted monopolar electrodes into the left prostate. (**c**) Sonographic images showing the identified prostate tissue (arrow) and the two monopolar electrodes (arrowheads).

**Figure 2 cancers-14-04987-f002:**
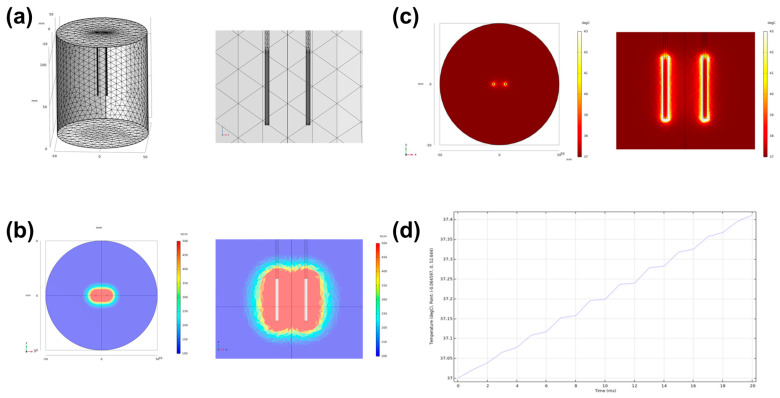
Electric field simulation in the prostate by high-frequency pulsed electric field using COMSOL Multiphysics 5.6. (**a**) Schematic image of monopolar electrodes being inserted into the simulated prostate. (**b**) The electric field strength at the axial- and longitudinal-sections between the two electrodes. (**c**) Thermal distribution at the axial- and longitudinal-sections when 1800 V/cm and 100 μs width of 10 pulses were applied between the two electrodes. (**d**) Temperature changes at the midpoint between the electrodes.

**Figure 3 cancers-14-04987-f003:**
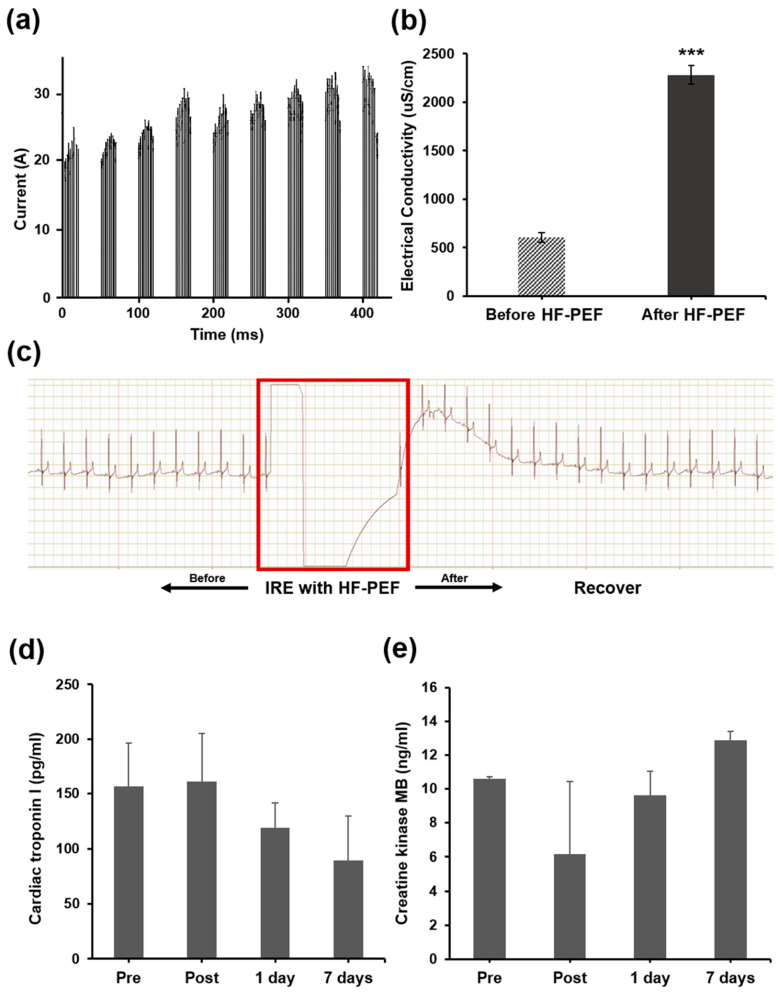
Current, electrical conductivity changes of the prostate tissue, cardiac safety, and biochemistry of the beagles after irreversible electroporation (IRE) with high-frequency pulsed electric field (HF-PEF). (**a**) Current of 9 groups of 10 pulses/group (interval times between pulses in the same group were omitted for being comparably long [~10 s]). (**b**) The electrical conductivity of prostate. Average treatment time was 420 ± 190 s. *** *p* < 0.001, compared with before IRE with HF-PEF. (**c**) Electrocardiogram of the beagle before and after IRE with HF-PEF (red box). Changes in (**d**) cardia troponin I and (**e**) creatine kinase MB levels.

**Figure 4 cancers-14-04987-f004:**
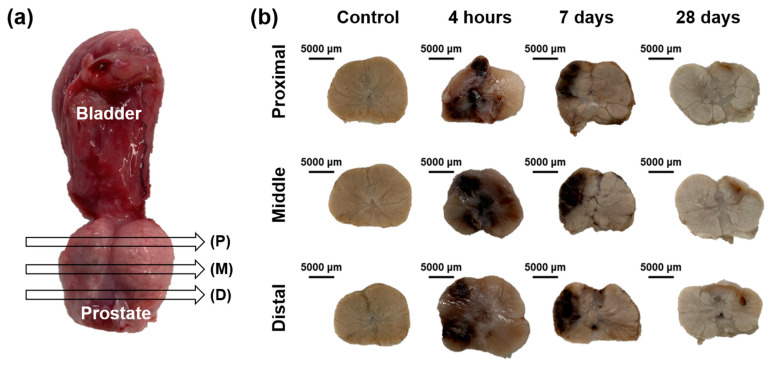
Tissue sampling and gross findings. (**a**) Prostate tissue samples showing the locations of (P) proximal, (M) middle, and (D) distal portions ablated by irreversible electroporation with high-frequency pulsed electric field. (**b**) Representative gross images of fixed prostate tissues.

**Figure 5 cancers-14-04987-f005:**
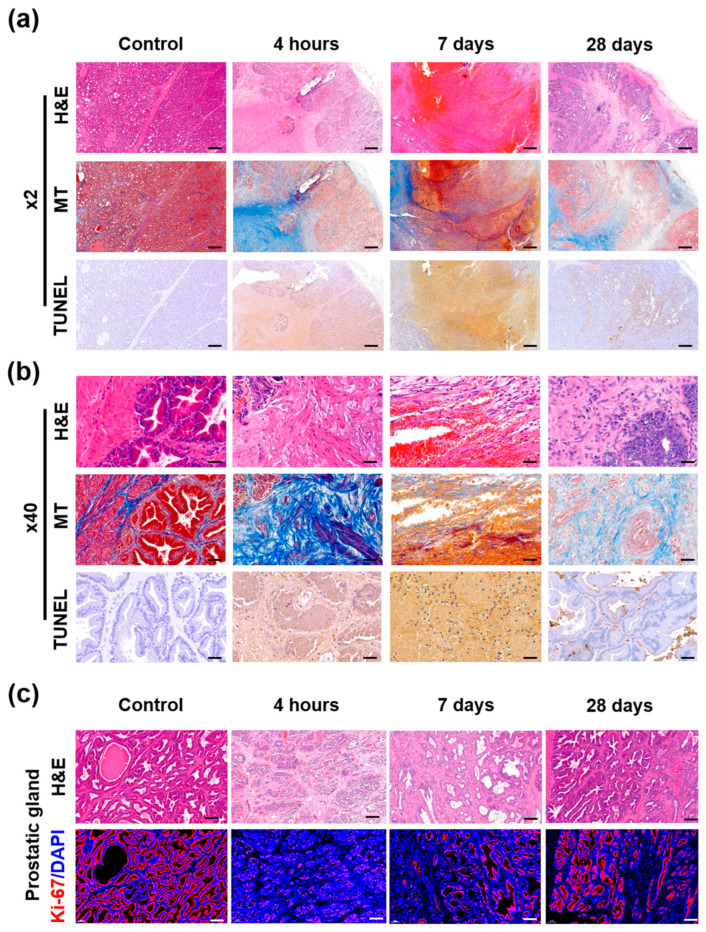
Representative microscopic images of the beagle prostate. Hematoxylin and eosin (H&E), Masson’s trichrome (MT), and terminal deoxynucleotidyl transferase-mediated dUTP nick and labeling (TUNEL)-stained sections shown at (**a**) 2× magnification (scale bar: 5 mm) and (**b**) 40× magnification (scale bar: 250 μm). (**c**) Ki-67 immunofluorescence (DAPI, blue; and Ki67, red) images showing active cellular regeneration (scale bar: 1 mm, 2× magnification).

**Figure 6 cancers-14-04987-f006:**
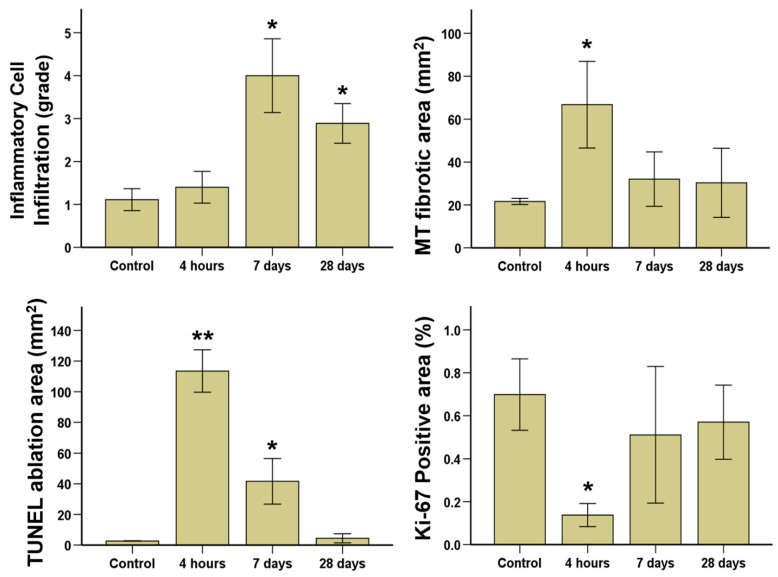
Histopathologic results of the control and treatment groups at different time points after irreversible electroporation with high-frequency pulsed electric field. * *p* < 0.05; ** *p* < 0.01, compared with control.

## Data Availability

Data generated or analyzed during the study are available from the corresponding author on request.

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
