# Peer review of "High-Frequency Pulsed Electric Field Ablation in Beagle Model for Treatment of Prostate Cancer"

_cancers, 2022, doi:10.3390/cancers14204987_

Round 1

Reviewer 1 Report

In the current paper Authors explored safety of feasibility of irreversibile electroporation by high frequency pulsed electric field in a beagle model.

The manuscript is extremely interesting, it is well written and flows well. I only have a few minor issues /edit suggestions to highlight

- In current literature there are already other preliminary experiences with HF-PEF in prostate model. Please have a look on DOI: 10.1115/1.405359. I would suggest to include this study within references and discuss it.

- Authors state in introduction "Possible mechanisms for frequency parameter of PEF are based on the high-frequency pass filter of the cellular membrane and transient rise of Ca2+ concentration in the cytosol". Indeed a previous study demonstrated that the cytotoxic effect is significantly enhanced in MHz compression of pulses and with the presence of extracellular Ca2+ ( DOI 10.1038/s41598-021-95180-7). It might be interesting in he very next future to explore such effects also on animal model. Please discuss. 

- Authors might consider to add few tables to summarize the main results

Author Response

Dear Editors of Cancers,

We present a revised manuscript for publication in Cancers, "High-Frequency Pulsed Electric Field Ablation in Beagle Model for Treatment of Prostate Cancer".

We thank each reviewer for the insightful suggestions directed to enhancing the quality of our work. In the following pages, we have provided all items revised according to the reviewers’ comments.

We hope that our revised version will be considered for publication in your journal. If you have any further questions about our manuscript, please do not hesitate to contact us by email. Thank you for your time and efforts that go into the publication of this paper.

Best regards,

Bumjin Lim M.D., Ph.D.1 and Jung-Hoon Park, Ph.D.2

1Department of Urology, Asan Medical Center, University of Ulsan College of Medicine, 88 Olympic-ro 43-gil, Songpa-gu, Seoul 05505, Republic of Korea

Tel: 82-2-3010-1835      Fax: 82-2-476-0090

E-mail: lbj1986@hanmail.net

2Biomedical Engineering Research Center, Asan Institute for Life Sciences, Asan Medical Center, 88 Olympic-ro 43-gil, Songpa-gu, Seoul, 05505, Republic of Korea

Tel: 82-2-3010-4123      Fax: 82-2-476-0090

E-mail: jhparkz@amc.seoul.kr

Response to the Reviewers’ comments

We would like to express our gratitude to the reviewers for their careful reading of the manuscript and for their insightful comments and suggestions. We have changed our manuscript in light of these remarks and believe that it has improved greatly. Our detailed replies to each reviewer comment are included below:

Reviewer 1

In the current paper Authors explored safety of feasibility of irreversibile electroporation by high frequency pulsed electric field in a beagle model. The manuscript is extremely interesting, it is well written and flows well. I only have a few minor issues /edit suggestions to highlight

1-1. In current literature there are already other preliminary experiences with HF-PEF in prostate model. Please have a look on DOI: 10.1115/1.405359. I would suggest to include this study within references and discuss it.

Response: Thank you for your comments. We have revised the Introduction and Discussion sections with additional reference as below.

In the line 61-66 on page 2 in the Introduction section

Meanwhile, High-frequency IRE (H-FIRE), biphasic pulses of 90~140 kHz, was designed to decellularize beagle prostate tissue for prostate cancer ablation (A1). However, relatively high voltages ranging 3,500~5,000 V were required to cause sufficient ablation area for H-FIRE. In this study, High-frequency PEF (HF-PEF) with monophasic pulses of 1.8 kV, 500 Hz was designed to ablate beagle prostate tissue.

In the line 277-280, 283 on page 12 in the Discussion section

In the recent, H-FIRE consisting of several bi-phasic pulses was proposed for prostate cancer ablation to reduce muscle contraction and prevent arrhythmias (A1). However, relatively high voltages ranging 3,500~5,000 V were required to cause sufficient ablation area for H-FIRE. Moreover, arrhythmias are less likely to occur because the prostate is located away from the heart (24). Despite previous studies on IRE with PEF, no studies have investigated the efficacy of IRE with HF-PEF consisting of several monophasic pulses in the beagle prostate, except for one study with a short period of observation (21).

A1. Aycock KN, Vadlamani RA, Jacobs EJ, Imran KM, Verbridge SS, Allen IC, Manuchehrabadi N, Davalos RV. Experimental and Numerical Investigation of Parameters Affecting High-Frequency Irreversible Electroporation for Prostate Cancer Ablation. J Biomech Eng. 2022 Jun 1;144(6):061003.

1-2. Authors state in introduction "Possible mechanisms for frequency parameter of PEF are based on the high-frequency pass filter of the cellular membrane and transient rise of Ca2+ concentration in the cytosol". Indeed a previous study demonstrated that the cytotoxic effect is significantly enhanced in MHz compression of pulses and with the presence of extracellular Ca2+ ( DOI 10.1038/s41598-021-95180-7). It might be interesting in the very next future to explore such effects also on animal model. Please discuss. 

Response: Thank you for your comments. We have revised the Introduction sections with additional references as below.

In the line 57-61 on page 2 in the Introduction section

In the recent in-vitro study, nanoseconds PEF (nsPEF) of 1 MHz frequency significantly enhanced cytotoxic effect of the prostate cancer cells in the presence of extracellular Ca2+ (A2). Despite of the enhanced cytotoxic effect in nsPEF, PEF of 1 kHz is estimated to cause less temperature rise than nsPEF, which causes thermal damage in the tissues (A3, A4).

A3. LackoviĆ I, Magjarević R, Miklavčič D, editors. A multiphysics model for studying the influence of pulse repetition frequency on tissue heating during electrochemotherapy. 4th European Conference of the International Federation for Medical and Biological Engineering; 2009: Springer.

A4. Mi Y, Rui S, Li C, Yao C, Xu J, Bian C, et al. Multi-parametric study of temperature and thermal damage of tumor exposed to high-frequency nanosecond-pulsed electric fields based on finite element simulation. Medical & biological engineering & computing. 2017;55(7):1109-22.

1-3. Authors might consider to add few tables to summarize the main results

Response: Thank you for your comments. We have added the table to summarize the main results in the supplementary table 1 as shown below.

We appreciate your thoughtful and informative review remarks. We are convinced that these comments improved the manuscript's quality greatly.

Reviewer 2 Report

The main concern is about the novelty of the content. Many other studies, as the authors have cited some, are working on IRE with higher frequencies. Authors claim that the low-frequency IRE should be used with larger electric fields with considerable side effects. Still, the electric field used by them has an even larger intensity than in previous studies. In short, I don't see any exciting novelty in the manuscript.

Some minor comments:

why does the current remain the same if the conductivity changes over time?

The simulation results could also be presented for other frequencies to indicate a high values limit for the number of steps and the frequency.

Author Response

Dear Editors of Cancers,

We present a revised manuscript for publication in Cancers, "High-Frequency Pulsed Electric Field Ablation in Beagle Model for Treatment of Prostate Cancer".

We thank each reviewer for the insightful suggestions directed to enhancing the quality of our work. In the following pages, we have provided all items revised according to the reviewers’ comments.

We hope that our revised version will be considered for publication in your journal. If you have any further questions about our manuscript, please do not hesitate to contact us by email. Thank you for your time and efforts that go into the publication of this paper.

Best regards,

Bumjin Lim M.D., Ph.D.1 and Jung-Hoon Park, Ph.D.2

1Department of Urology, Asan Medical Center, University of Ulsan College of Medicine, 88 Olympic-ro 43-gil, Songpa-gu, Seoul 05505, Republic of Korea

Tel: 82-2-3010-1835      Fax: 82-2-476-0090

E-mail: lbj1986@hanmail.net

2Biomedical Engineering Research Center, Asan Institute for Life Sciences, Asan Medical Center, 88 Olympic-ro 43-gil, Songpa-gu, Seoul, 05505, Republic of Korea

Tel: 82-2-3010-4123      Fax: 82-2-476-0090

E-mail: jhparkz@amc.seoul.kr

Response to the Reviewers’ comments

We would like to express our gratitude to the reviewers for their careful reading of the manuscript and for their insightful comments and suggestions. We have changed our manuscript in light of these remarks and believe that it has improved greatly. Our detailed replies to each reviewer comment are included below:

Reviewer 2

2-1. The main concern is about the novelty of the content. Many other studies, as the authors have cited some, are working on IRE with higher frequencies. Authors claim that the low-frequency IRE should be used with larger electric fields with considerable side effects. Still, the electric field used by them has an even larger intensity than in previous studies. In short, I don't see any exciting novelty in the manuscript.

Response: Thank you for your comments. The HF-PEF in our study was performed under following parameters: electric field strength, 1,800 V/cm; pulse width, 100 μs; and pulse interval, 2 ms. A total of 90 pulses were divided into 9 cycles of 10 pulses each to avoid thermal damage from intense electrical current. We expected from our current study that the parameters used for the HF-PEF provide reducing the number of muscle contractions during the procedure and shortening the procedure time. In the best knowledge, there are no study to investigate the efficacy of IRE with HF-PEF consisting of several monophasic pulses in the beagle prostate. We have added the sentences with additional references in the Introduction and Discussion sections as below.

In the line 57-61 on page 2 in the Introduction section

In the recent in-vitro study, nanoseconds PEF (nsPEF) of 1 MHz frequency significantly enhanced cytotoxic effect of the prostate cancer cells in the presence of extracellular Ca2+ (A2). Despite of the enhanced cytotoxic effect in nsPEF, PEF of 1 kHz is estimated to cause less temperature rise than nsPEF, which causes thermal damage in the tissues (A3, A4).

A3. LackoviĆ I, Magjarević R, Miklavčič D, editors. A multiphysics model for studying the influence of pulse repetition frequency on tissue heating during electrochemotherapy. 4th European Conference of the International Federation for Medical and Biological Engineering; 2009: Springer.

A4. Mi Y, Rui S, Li C, Yao C, Xu J, Bian C, et al. Multi-parametric study of temperature and thermal damage of tumor exposed to high-frequency nanosecond-pulsed electric fields based on finite element simulation. Medical & biological engineering & computing. 2017;55(7):1109-22.

In the line 277-280, 283 on page 12 in the Discussion section

In the recent, H-FIRE consisting of several bi-phasic pulses was proposed for prostate cancer ablation to reduce muscle contraction and prevent arrhythmias (A1). However, relatively high voltages ranging 3,500~5,000 V were required to cause sufficient ablation area for H-FIRE. Moreover, arrhythmias are less likely to occur because the prostate is located away from the heart (24). Despite previous studies on IRE with PEF, no studies have investigated the efficacy of IRE with HF-PEF consisting of several monophasic pulses in the beagle prostate, except for one study with a short period of observation (21).

A1. Aycock KN, Vadlamani RA, Jacobs EJ, Imran KM, Verbridge SS, Allen IC, Manuchehrabadi N, Davalos RV. Experimental and Numerical Investigation of Parameters Affecting High-Frequency Irreversible Electroporation for Prostate Cancer Ablation. J Biomech Eng. 2022 Jun 1;144(6):061003.

H-FIRE requires even more higher electric field intensity of 3,500~5,000 V to ablate the beagle prostate tissue. HF-PEF was designed to have lower electric field intensity than H-FIRE protocol and higher frequency than conventional PEF protocol to get the advantages of both previous protocols; safety and efficacy as shown below.

Some minor comments:

2-2. why does the current remain the same if the conductivity changes over time?

Response: Thank you for your comments. As shown in Figure 3A, the current gradually increased over time. We have changed the Figure 3A to show more dramatic current changes over time. The inhomogeneous characteristics of the tissues or temperature affect the current changes.

2-3. The simulation results could also be presented for other frequencies to indicate a high values limit for the number of steps and the frequency.

Response: Thank you for your comments and we have agreed your comments. Unfortunately, only the parameter used the HF-PEF in the prostate was simulated using COMSOL Multiphysics 5.6.

We appreciate your thoughtful and informative review remarks. We are convinced that these comments improved the manuscript's quality greatly.

Round 2

Reviewer 2 Report

Considering the revisions, I think the manuscript can be published in its current form.